# Isolation of Mirificin and Other Bioactive Isoflavone Glycosides from the Kudzu Root Lyophilisate Using Centrifugal Partition and Flash Chromatographic Techniques

**DOI:** 10.3390/molecules27196227

**Published:** 2022-09-22

**Authors:** Magdalena Maciejewska-Turska, Łukasz Pecio, Grażyna Zgórka

**Affiliations:** 1Department of Pharmacognosy with the Medicinal Plant Garden, Faculty of Pharmacy, Medical University of Lublin, 1 Chodzki Str., 20-093 Lublin, Poland; 2Department of Biochemistry and Crop Quality, Institute of Soil Science and Plant Cultivation, State Research Institute, 24-100 Pulawy, Poland

**Keywords:** kudzu root, mirificin, isoflavone glycosides, centrifugal partition chromatography, flash chromatography

## Abstract

*Pueraria lobata* (Willd.) Ohwi is a legume taxon native to Southeast Asia and widely used in traditional medicine systems of that region. The therapeutic applications of the underground parts of this species (known as kudzu root) are related to its high content of isoflavones, mainly the characteristic *C*-glycoside derivatives. Within this group, the most explored compound both phytochemically and pharmacologically is puerarin. However, current scientific findings document important anti-biodegenerative effects for some of the minor isoflavones from kudzu roots. Therefore, the main objective of the study was to develop an original preparative method that allowed the efficient isolation of closely related hydrophilic daidzein *C*-glycosides, including mirificin, from vacuum-dried aqueous-ethanolic extracts of kudzu roots. For this purpose, the combined centrifugal partition (CPC) and flash chromatographic (FC) techniques were elaborated and used. The optimized biphasic solvent system in CPC, with ethyl acetate, ethanol, water, and 0.5% (*V/V*) acetic acid as a mobile phase modifier, enabled the purification and separation of the polar fraction containing bioactive isoflavones and ultimately the isolation of mirificin, 3′-hydroxy- and 3′-methoxypuerarin, puerarin, and daidzin using FC. The identity of isoflavones was confirmed using spectroscopic (UV absorption and nuclear magnetic resonance) and mass spectrometric methods. The determined purity of isolated mirificin was 63%.

## 1. Introduction

*Pueraria lobata* (Willd.) Ohwi, known as kudzu or Gegen, is a perennial, creeping vine belonging to the family Fabaceace [1]. This highly invasive plant was one of the oldest herbal substances used in traditional Chinese edicine to treat the common cold and diarrhea, and today kudzu root, listed in the Chinese Pharmacopoeia as an agent indicated in the complementary therapy of hypertension and angina pectoris, is used in various forms of pharmaceutical formulations prepared thereof [2,3,4]. The monograph of kudzuvine root included in the European Pharmacopoeia describes the qualitative criteria for this herbal substance concerning the content of total isoflavonoids [5]. In this group, for many years, puerarin (daidzein-8-*C*-glucoside) has been considered the predominant bioactive component exhibiting selective estrogen receptor modulating (SERM) effects [6,7,8]. The ability to counteract estrogen deficiency demonstrated by this isoflavone has resulted in the use of kudzu preparations in the treatment of age-related disorders especially in postmenopausal women, involving atherosclerotic cardiovascular disease [2,9], neurological dysfunction [10], osteoporosis [11,12], and metabolic syndrome resulting in obesity and diabetes [13,14]. Puerarin and kudzu extracts are also known as natural antidipsotropic agents recommended for alcohol abuse and alleviating withdrawal syndrome [15,16]. Current phytochemical studies on *P. lobata* have led to identification of some related daidzein glycosides, such as 3′-hydroxy- and 3′-methoxypuerarin, 6-*O*”-xylosylpuerarin, daidzin, and mirificin, which, together with formononetin and its 7-*O*-glycoside (ononin) may be responsible for the synergetic pharmacological effects attributed to kudzu [17,18,19,20]. Therefore, in the present study, we focused on these little-known kudzu isoflavones, specifically daidzein-8-*C*-apiosyl-(1→6)-glucoside, commonly referred to as puerarin apioside or mirificin, considering some published reports (described below) on the interesting biological effects of this plant metabolite. In 2016, Xiao et al. [21] documented that mirificin could easily cross the blood–brain barrier after intravenous administration of *P. lobata* isoflavones (80 mg/kg) in rats, which may have been related to the neurotrophic effects later described for kudzu root [22]. In other studies, mirificin, 3′-hydroxy- and 3′-methoxypuerarin, and puerarin dose-dependently affected the levels of striatal neurotransmitters, mainly dopamine (DA), serotonin (5-HT), and glutamate (Glu) in rats. After intravenous administration of a higher dose of isoflavones (160 mg/kg), a decrease in DA concentration and inhibition of 5-HT metabolism were observed. In addition, a significant neuroprotective effect of isoflavones has been documented due to a decrease in extracellular Glu levels, which may prevent neurocyte damage resulting from Glu excitotoxicity under ischemia/stroke conditions [10]. Mirificin was also proved to be a potent tyrosinase inhibitor *in vitro*, showing a stronger pharmacological effect than kojic acid [23]. Other researchers also confirmed that the location of glucose moiety in the molecular structure of isoflavone *C*-glycosides had a significant impact on their biological properties. As regards to mirificin, the results of molecular docking showed that this compound could serve as a potential structural target with high affinity towards the catalytic region of tyrosinase that determined its potential in melanoma and skin tumor prevention [24]. The noticeable increase in research on the pharmacological effects of mirificin has not been accompanied by parallel progress in phytochemical investigations related to the efficient isolation of this bioactive component on a preparative scale.

To date, various strategies have been elaborated to separate isoflavones from complex plant matrices [25,26]. Since conventional methods were tedious, requiring much time, high consumption of solvents [20], or numerous experimental steps [27], other efficient preparative procedures started to be developed, especially in the last decade, including high-speed (HSCCC) or, more advanced technologically, high-performance (HPCCC) counter-current chromatography [19,28]. These hydrodynamic separation techniques are based on the continuous partitioning of molecules between two immiscible liquid phases which prevents irreversible adsorption of sample components observed in traditional column chromatographic methods [29]. The original idea of semi-preparative isolation of some puerarin derivatives from *P. lobata* extract using HSCCC was proposed in 1999 by Cao and colleagues [30]. Further improvements to the method were made by Li et al. [19] who obtained daidzein and its four glycosides (daidzin, puerarin, 3′-hydroxypuerarin, and 3′-methoxypuerarin) using HPCCC and *n*-hexane-ethyl acetate-*n*-butanol-ethanol-water (0.5:2:1:0.5:3.5, *V*/*V*/*V*/*V*/*V*) as a biphasic solvent system. To date, there has been only one scientific report, published by Sun et al. [31], on the use of fast centrifugal partition chromatography (FCPC) for the micropreparative isolation of the predominant compound (puerarin) of a dry extract of *P. lobata* using a mobile/stationary phase system consisting of ethyl acetate-*n*-butanol-water in a 2:1:3 (*V/V/V*) ratio. The researchers also confirmed the superiority of FCPC over the comparatively used HSCCC in the one-step isolation and purification of puerarin, resulting in better resolution of the other components of the kudzu extract and higher purity of the aforementioned isoflavone glucoside.

In this study, we proposed an original two-step chromatographic approach for the efficient and simple preparative isolation of some minor isoflavone glycosides, including mirificin, from a vacuum-dried aqueous-ethanolic extract of kudzu roots. In the first step, hydrostatic centrifugal partition chromatography under controlled pH conditions was developed to purify and concentrate the polar fraction of isoflavones, followed by flash chromatography, which was used (in the second step) to properly separate closely related compounds and ultimately to obtain mirificin and other related isoflavone glycosides of high purity.

## 2. Results and Discussion

### 2.1. Qualitative and Quantitative Analysis of Isoflavone Constituents of the Kudzu Root Lyophilisate

The initial step of the preparative studies was to do a detailed qualitative and quantitative profiling of the kudzu root lyophilisate using reversed-phase liquid chromatography (RP-LC) with PDA detection, followed by mass spectrometric analysis performed under the conditions described below (Section 3.5 and Section 3.6). The results of this analysis revealed the presence of seven isoflavone components, including the five polar glycosides (compounds **1–5**) shown in Figure 1. The established concentration of puerarin, as the predominant constituent of the kudzu root lyophilisate, was 17.1%, calculated for a dry mass of the herbal preparation. The content of the other four polar isoflavones (compounds **1** and **3–5**) was 0.7, 3.9, 4.6, and 5.4%, respectively. The quantities of ononin (compound **6**) and formononetin (compound **7**) were established as 0.3 and 2.5% of dry lyophilisate mass, respectively. Mirificin (**4**), as the main bioactive glycoside of interest, had very similar retention on octadecylsilane stationary phase compared to puerarin (**2**) and 3′-methoxypuerarin (**3**). Therefore, obtaining pure forms of the aforementioned isoflavone isolates and removing potential non-phenolic ballast compounds from the kudzu lyophilisate using preparative RP-LC as the only separation method was found to be inefficient.

An additional problem in isolating the above isoflavone components was the predominant content of puerarin in the plant preparation that was about five-fold higher compared to 3′-methoxypuerarin, mirificin, and daidzin, and even more than ten times higher in relation to 3′-hydroxypuerarin. While analyzing the molecular structure of the aforementioned compounds (Figure 2), we noticed that, except for daidzin, the other isoflavones have free phenolic groups at C-7 and C-4’, which indicated the possibility of steering their retention in liquid biphasic systems using electron-donor components such as ethyl acetate.

Therefore, in the first stage of the preparative studies, we decided to introduce an original scientific approach by using centrifugal partition chromatography (CPC), which provided the opportunity to exclusively purify, concentrate, and isolate the polar fraction of isoflavone glycosides from the kudzu root lyophilisate, based on the polyphenolic nature of these compounds.

### 2.2. Purification and Concentration of the Polar Fraction of Isoflavones Using CPC

#### 2.2.1. Selection and Optimization of the Biphasic Solvent System in CPC

Several biphasic solvent systems have previously been used by other authors to separate kudzu isoflavone components using HSCCC [32,33,34,35]. However, the hydrodynamic retention mechanism in HSCCC caused by the varying gravitational field created by the planetary motion of the rotors differs significantly from the hydrostatic CPC, which relies on the constant gravitational interactions observed during the motion of a uniaxial rotor [36]. The differences between the two methods mentioned above necessitated our own step-by-step optimization of the separation conditions, based on the technical parameters of the CPC apparatus and the expected practical effects of the preparative procedure. In selecting the eluents, we had also to take into account the detailed reports on phytochemicals identified in kudzu root extracts [2,37], which indicated the presence of various non-chromophore components, such as saponins, sterols, or lignans that could interfere with obtaining pure isoflavone isolates. In order to achieve such a final result, it was necessary to determine the correct partition coefficient (*K_D_*) for puerarin, which served as a model isoflavone glycoside for selecting the optimal biphasic solvent system. The suitable *K_D_* values (in the range of 0.5–2.0) for counter-current chromatography were established by Ito [38]. Referring to the successful results of separation and purification of puerarin by FCPC in the ethyl acetate-*n*-butanol-water system published by Sun et al. [31], we first decided to test this solvent composition using different volume ratios of the individual components (Table 1).

Unfortunately, *K_D_* values above 1.35 were obtained in all butanol biphasic systems examined (Table 1**,** No **1–3**), indicating much stronger retention of puerarin in the non-polar upper phase. Therefore, these solvent compositions were completely unsuitable for our preparative purposes, as they would have prevented the separation of the polar isoflavone fraction from the hydrophobic ballast compounds. In the next step, we made a swap for the more polar aliphatic alcohol-ethanol-in the solvent systems examined, which resulted in a reduction of the *K_D_* value to about 1.0 (Table 1**,** No **4–5**). Since this was still unsatisfactory, in the last optimization step we decided to use acetic acid as the pH modifier and molecule dissociation regulator in a two-phase solvent system. This modification resulted in the effective distribution of puerarin into the lower polar phase, as evidenced by *K_D_* values below 0.5 (Table 1**,** No **6–7**). Ultimately, the solvent system EtOAc-EtOH-H_2_O/10:1:10 (*V/V/V*) with the addition of 0.5% (*V/V*) CH_3_COOH was selected as the optimal one for CPC isolation of the polar isoflavone fraction using the lower phase as a mobile in the descending mode.

#### 2.2.2. Effectiveness of the CPC Method in Purification and Isolation of Polar Fraction of Kudzu Isoflavone Glycosides

Due to the addition of an acidic modifier to the solvent system, a significant improvement in the separation efficiency of the lyophilisate components was observed during the CPC run. Lowering the pH in the mobile phase resulted in inhibition of dissociation of phenolic and carboxyl/hydroxyl substituents in the molecules of kudzu lignans and sapogenins/saponins, respectively, followed by more intense migration of these ballast compounds to the upper (stationary) phase. On the other hand, the optimized *K_D_* value for the solutes of interest resulted in a high distribution of the polar puerarin and daidzein derivatives into the lower (mobile) phase (Figure 3).

A rectangular, symmetrical peak shape was recorded for the entire isoflavone fraction, as is usually observed in the displacement chromatography model (Figure 3). Similar peak profiles were previously described by Ito and Ma [39] for separation processes of various compounds using pH-zone-refining CCC, based on controlled pH in both the stationary and mobile phases. In our study, the addition of an appropriate volume of acetic acid lowered the pH in the biphasic solvent system and made it possible not only to remove more hydrophobic ballast compounds, retained in the stationary phase, but also to reduce the ionization of free phenolic groups in the molecules of mirificin and related polar isoflavone glycosides (Figure 2). Ultimately, this led to efficient pre-concentration and subsequent rapid elution of the entire fraction of polar isoflavones between 70 and 75 min (Figure 3).

The target compounds were monitored with a UV detector at 260 nm. All collected fractions of interest were combined and evaporated to dryness and then subjected to further separation using flash chromatography. It is noteworthy that the use of the lower part of the acidified EtOAc-EtOH-H_2_O solvent system as the mobile phase provided the additional benefit of removing polysaccharide ballast compounds from the lyophilisate during the first 60 min of the CPC run. This was an important achievement, as kudzu root contains high amounts of starch (about 15–30% of the fresh plant material), which is extracted from this herbal substance and used as a dietary and nutritional agent in several Asian countries [40]. However, in the process of preparative isolation of isoflavones, the aforementioned polysaccharide component constituted a substantial ballast that made it difficult to obtain pure crystalline fractions of these compounds when separated using the same polar eluent in the descending mode.

### 2.3. Isolation of Mirificin and Related Polar Isoflavone Glycosides Using Flash Chromatography

The second part of the preparative studies conducted was devoted to the isolation of mirificin and the remaining polar isoflavone glycosides from the combined fraction obtained by the CPC method. For this purpose, a reversed-phase flash chromatography (FC) technique was employed. An important methodological improvement that increased the separation efficiency of this method was the use of the Samplet^TM^ cartridge, which provided significant pre-concentration of chromatographic zones (Figure 4). On the other hand, the optimized gradient profile of the eluent consisting of 1% (*V/V*) acetic acid with a slowly increasing participation of an alcoholic modifier (methanol) enabled further purification of isoflavone fraction from hydrophilic polysaccharide compounds (during the first 40 min of the FC run) that were removed to waste. A further increase (from 16 to 26%) in the volume proportion of methanol in the mobile phase yielded the main fractions of interest (from 64 to 115), which contained the separated isoflavone compounds (Figure 4). After the control RP-LC/PDA analysis, which confirmed that the retention times and UV spectra of the contained single isoflavone components were consistent with those documented for the reference samples, the collected FC eluates: 64–73, 77–86, and 87–99 (Figure 4) were combined and concentrated to dryness, and finally labeled as fraction I (with 3′-hydroxypuerarin), II (containing puerarin), and III (3′-methoxypuerarin), respectively.

As for the combined eluates 105–115, after condensing to about 50 mL and cooling at 4 °C, two sub-fractions were obtained: IVa (precipitate of mirificin), and IVb (supernatant containing daidzin), which were separated, and the liquid fraction was re-concentrated under vacuum. Finally, all crystalline isoflavone isolates, obtained in the five concentrated fractions I-III, IVa, and IVb, were numbered as compounds **1–5**, respectively, and further examined to confirm their chemical identity using chromatographic, spectroscopic, and spectrometric methods described in Section 2.5.

### 2.4. Graphical Presentation of Preparative Extraction and Chromatographic Procedures

To better illustrate the detailed protocol developed for preparative-scale isolation of kudzu root constituents, Figure 5 schematically shows all the techniques used on the way from plant material to bioactive isoflavone components.

### 2.5. Molecular Structure Analysis of the Obtained Isoflavone Isolates Using Advanced Spectrometric and Spectroscopic Techniques

A final step of our study was the assessment and/or confirmation of the detailed chemical structure of obtained isoflavone isolates. For this purpose, high-performance liquid chromatography (LC), hyphenated with PDA detection (that enabled UV spectra acquisition), and tandem mass spectrometry (ESI-NEG-QToF/MS-MS) with electrospray (ESI) ionization in negative polarization mode (NEG) was used (Agilent Technologies, Santa Clara, CA, USA). The MS/MS system consisting of a coupled quadrupole (Q) and time-of-flight (TOF) mass analyzers and equipped with MassHunter software (Agilent Technologies) provided high-resolution spectra acquisition with deprotonated molecular and fragment ions for all isoflavone isolates. The identity of puerarin (**2**), as the predominant compound isolated from kudzu lyophilisate, was initially confirmed by comparing its chromatographic and UV spectral data with those obtained for the authentic substance. However, a rich MS/MS spectrometric data set provided the opportunity to trace in detail the fragmentation pattern of this compound and other related 8-*C*-isoflavone glycosides obtained from this herbal preparation. Therefore, we additionally carried out our own theoretical studies on the fragmentation pathways of the aforementioned isoflavone glycoconjugates (especially compound **4**, supposed to be mirificin), using MS/MS spectra generated from deprotonated molecular ions. Based on the literature data [41], we found that unlike the fragmentation of *O*-glycosidic flavonoid derivatives, which lose the entire sugar moiety, this process was not observed for *C*-glycosides, due to the difficulty in breaking the C-C bond between the monosaccharide and the aglycone structure. As documented by other authors [42,43], under the influence of the collision-induced dissociation (CID) energy of −20 and −40 eV, the hexose ring, attached by a C-C bond to the isoflavone skeleton at the *C*-8 position, underwent a characteristic cleavage of the 0/2 and 0/3 bonds causing a decrease in *m*/*z* values by 120 Da and 90 Da, respectively. Referring to the mass spectrometric analysis of compound **4**, the generated precursor ion of this component at *m*/*z* 547.1495 was consistent with the molecular formula C_26_H_28_O_13_ and the hypothetical structure of isoflavone 8-*C*-glycoside with a two-component sugar chain, which was proposed by the MassHunter chemical formula generator. In addition, a fragment ion with *m/z* value of 415.1089 indicated neutral elimination of the entire apioside moiety (132 Da) from the deprotonated molecular ion (Figure 6). A similar abundant molecular ion [M − H]^−^ at *m*/*z* 415.1077 and fragmentation pattern was obtained earlier for compound **2** unambiguously identified as puerarin [44].

The diagnostic fragment ions obtained at *m/z* 325.0765 and 295.0644 for this component, which confirmed the structure of the isoflavone *C*-glycoside, were also consistent with those obtained for puerarin. Accordingly, compound **4** was tentatively identified as mirificin, and the proposed fragmentation pattern has been shown in Figure 5. Other polar isoflavone isolates, marked as compounds **1** and **3**, yielded molecular ions at *m/z* 431.1018 and 445.1089, respectively. Further, MS/MS fragmentation resulted in abundant product ions at *m/z* 415, for both isoflavones analyzed, which indicated the presence of an additional hydroxyl group (16 Da) in the structure of compound **1**, and a methoxy group (30 Da) in compound **3**, compared to puerarin, and allowed their tentative identification as 3′-hydroxypuerarin and 3′-methoxypuerarin, respectively. For the last isoflavone isolate examined (compound **5**), an adduct ion [M + HCOO]^−^ was obtained at *m/z* 461.1132 and a precursor ion at *m/z* 415.1087, which in further fragmentation formed a product ion at *m/z* 253.0522, resulting from a simple C-O bond cleavage between a sugar (glucose) and an aglycone (daidzein) structure. Therefore, this compound, the identity of which was previously confirmed using a reference substance (retention time and UV spectrum agreement in RP-LC/PDA analysis) was identified as daidzein.

As regards isoflavone isolates, in total, 68 mg of puerarin, 2 mg of 3′-hydroxypuerarin, 15 mg of 3′-methoxypuerarin, 18 mg of mirificin, and 20 mg of daidzin were obtained using all combined preparative procedures. The correctness of the molecular structure assessment by mass spectrometry for four isoflavone isolates (compounds **2–4**), which were obtained in sufficient quantities (more than 5 mg), was undoubtedly confirmed by the detailed NMR analysis, carried out under the conditions described in Section 3.6. The results obtained, as presented in the attached Appendix A (Appendix A), were consistent with those obtained previously by other researchers [45,46,47]. In addition, the NMR method (ERETIC2), using external standard (3,5-dihydroxybenzoic acid), made it possible to evaluate the purity of individual isoflavone isolates, including puerarin (95%), 3′-methoxypuerarin (80%), mirificin (63%), and daidzin (75%). For comparison, we have presented in Table 2 the purity values of the same kudzu root isoflavones isolated by other researchers using counter-current preparative techniques. However, as can be seen in the first column of Table 2, mirificin was not obtained in any of these studies.

## 3. Materials and Methods

### 3.1. Solvents and Other Reagents

The analytical purity solvents (ethanol, methanol, butanol, ethyl acetate, and acetic acid) used for preparative procedures were purchased from Avantor Performance Materials (Gliwice, Poland). Purified water (18.2 MΩ) was obtained using a Direct-Q system (Millipore, Molsheim, France). Reagents of chromatographic or spectroscopic purity (acetonitrile, methanol, formic acid, and water) used for the qualitative and quantitative profiling of kudzu lyophilisate were provided by J.T. Baker (Gross-Gerau, Germany). Puerarin (>95% purity) was purchased from ChromaDex Inc. (Santa Ana, CA, USA). Dimethyl sulfoxide-d6, 99.8 atom %D (DMSO-*d_6_*) and 0.1% trifluoroacetic acid (TFA), used for recording proton nuclear magnetic resonance (^1^H-NMR) spectra, were provided by Merck (Darmstadt, Germany).

### 3.2. Plant Material and Obtaining a Vacuum-Dried Kudzu Root Extract

The roots of *P. lobata*, sourced from China, had a certificate of authenticity of the herbal substance and were supplied in the form of crushed and dried pieces by a Polish distributor (NatVita, Poznań). Before extraction, the plant material was additionally dried at 35°C for 10 h and pulverized in a Pulverisette 19 cutting mill (Fritsch GmbH, Idar-Oberstein, Germany), equipped with a 1 mm sieve. The powdered kudzu root sample (30 g) was then introduced into a 1000 mL round flask and 500 mL of 50% (*V/V*) ethanol was added, followed by ultrasound-assisted extraction and vacuum drying procedures under conditions previously described by Maciejewska-Turska and Zgórka [48]. The resulting lyophilisate (~8.250 g), after preliminary qualitative and quantitative profiling (Section 3.5), was subjected to further preparative chromatographic procedures.

### 3.3. Centrifugal Partition Chromatography (CPC)

#### 3.3.1. Determination of Partition Coefficients for Various Solvent Systems

Due to the specific equilibrium separation process in CPC, partition coefficients (*K_D_*) were first determined, which represented the ratio of analyte concentration in the liquid upper and lower phases obtained by a series of liquid-liquid extractions carried out in separatory funnels for a reference isoflavone substance (puerarin) using different solvent systems. Several biphasic solvent systems were used, consisting of ethyl acetate, aliphatic alcohols (butanol or ethanol), and water, in various volumetric ratios, with (or without) acetic acid as a pH modifier (Table 1). For *K_D_* determination, 2 mg of puerarin was added to a separatory funnel and shaken for 3 min with a mixture of the upper (5 mL) and lower phase (5 mL), which were prepared previously for each solvent system examined. After the two phases reached equilibrium, 1 mL of each phase was pipetted, concentrated to dryness under vacuum, and analyzed for the concentration of puerarin using the RP-HPLC/PDA method (Section 3.5).

#### 3.3.2. Apparatus and Chromatographic Procedure

Centrifugal partition chromatography was performed using an Armen SCPC-250-L system (Saint Ave, France) equipped with a quaternary pump, a 250 mL rotor, a UV detector, and a fraction collector. An optimized biphasic solvent system: ethyl acetate-ethanol-water (10:1:10, *V/V/V*) with 0.5% (*V/V*) acetic acid, was prepared in a glass separatory funnel to obtain appropriate volumes (1 L) of both the upper and lower phases. The CPC process was started by filling the column with the upper phase (stationary phase) at 500 rpm and a flow rate of 20 mL/min for 25 min. The lower phase (serving as the mobile phase) was then pumped until hydrostatic equilibrium was reached. Subsequently, 100 mg of kudzu lyophilisate was dissolved in a 1:1 (*V/V*) mixture of lower and upper phases (5 mL) and injected into the system using a Rheodyne injector with a 10-mL sampling loop. The entire chromatographic separation was performed in descending mode at 1000 rpm and a flow rate of 3 mL/min for 95 min. Eluates were monitored with a UV detector at two wavelengths: 260 nm (isoflavonoids) and 320 nm (lignans) and collected into glass tubes placed in a fraction collector. The CPC procedure was performed five times. The same fractions of interest containing polar isoflavonoids were combined and concentrated under vacuum, and the total dry residue (~180 mg) was obtained for further separation using flash chromatography (Section 3.4).

### 3.4. Flash Chromatography (FC)

The polar fraction of isoflavone glycosides obtained by the CPC method (Section 3.3.2) was further separated using an Isolera One flash chromatography apparatus (Biotage, Uppsala, Sweden) combined with a fraction collector and a UV detector. FC procedure was carried out on a Biotage SNAP KP-C18-HS reversed-phase cartridge (60 g, d_p_ = 50 μm), which was conditioned with a mobile phase (starting composition described below) using at least three column volumes. In addition, a self-packed Samplet^TM^ cartridge (12 g), containing the same octadecyl adsorbent, was used to load a sample (~50 mg) of dry isoflavone residue obtained by the CPC method. Prior to that, the sample was dissolved in 20 mL of 50% (*V/V*) methanol and applied (in several portions) to the surface of the adsorbent bed using a pipette. After each portion was loaded, the samplet cartridge was placed in a dryer to evaporate the solvent at 50 °C. Finally, the cartridge was inserted into the SNAP column and the FC process was run. To separate the sample components, a mobile phase consisting of 1% (*V/V*) acetic acid (solvent A) and methanol (solvent B) was used at a constant flow rate of 25 mL/min according to the preliminary isocratic program: 3%B (0–10 min), followed by a linear increase to 30%B (at 95 min) and to 45%B in A (at 155 min). Detection of compounds was carried out at 260 nm. The whole FC procedure was carried out thrice. The purity and content of individual isoflavone isolates in combined and condensed eluates were evaluated using the RP-LC/PDA method as described in Section 3.5.

### 3.5. RP-LC/PDA Protocol

The reversed-phase liquid chromatography (RP-LC) was used to determine the qualitative and quantitative profile of the kudzu root lyophilisate, the obtained CPC and FC fractions, and the identity and purity of the final isoflavone isolates. The chromatographic system consisted of an Agilent Technologies 1100 series chromatograph (Waldbronn, Germany) equipped with a Rheodyne injector, a 20-μL sample loop, and a photodiode array (PDA) detector. Separation of isoflavone components was carried out on an Aquasil C18 column (250 × 4.6 mm I.D., d_p_ = 5 μm) at 25 °C using a gradient of component A (1 mM H_3_PO_4_, pH~3) and B (acetonitrile) as follows: 0/10; 25/10; 60/30; 80/95; 85/10 min/%B in A and a flow rate of 1 mL/min. Post time was set at 10 min. UV spectra were recorded at 260 nm (isoflavonoids) and 210 nm (tracing impurities). The content of isoflavones in the kudzu lyophilisate and all CPC and FC fractions obtained was calculated as puerarin. For this purpose, a set of six methanolic solutions of this reference substance at concentrations ranging from 0.01 to 0.15 mg/mL was prepared and used to determine the calibration curve and linear regression equation.

### 3.6. Mass Spectrometry (MS) and Nuclear Magnetic Resonance (NMR)

An Agilent 1260 Infinity LC chromatograph coupled to a photodiode array detector and 6530 Accurate Mass Q-TOF LC/MS system (Agilent Technologies, Santa Clara, CA, USA) was used to confirm the identity of isoflavonoid isolates. Separation of isoflavonoid components was performed on a Zorbax SB-C18 narrow-bore column (2.1 × 150 mm, d_p_ = 3.5 µm). A gradient of acetonitrile (B) and water (A), together with eluent modifiers such as 0.1% (*V/V*) formic acid and 10 mM ammonium formate, was used according to a previously described LC-MS procedure [48]. Electrospray ionization (ESI) in negative ion mode (NEG) was used to generate isoflavone ions in the MS system. Spectral data acquisition was performed using the Agilent MassHunter Workstation. Confirmation of the chemical structure of the analyzed isolates was carried out on the basis of the obtained *m/z* values for molecular and fragmentation ions compared with data available in scientific MS databases and the self-analysis of potential fragmentation pathways.

The nuclear magnetic resonance (NMR)-spectra—^1^H, ^13^C, 1D-ROESY (rotating-frame Overhauser effect spectroscopy), COSY (correlation spectroscopy), TOCSY (total correlation spectroscopy), HSQC (heteronuclear single quantum coherence), H2BC (heteronuclear two-bond coherence), and HMBC (heteronuclear multiple bond coherence)—of isoflavonoid isolates were recorded at 30 °C in DMSO-d6 (99.8%D) with 0.1% trifluoroacetic acid. The spectra were calibrated to the tetramethylsilane signal and recorded using standard pulse programs on an Avance III HD Ascend 500 MHz spectrometer (Bruker BioSpin, Rheinstetten, Germany).

## 4. Conclusions

This paper presents original and comprehensive research focused on the efficient isolation of the polar fraction of bioactive isoflavone glycosides (mirificin, 3′-hydroxypuerarin, 3′-methoxypuerarin, and daidzin), in the presence of predominant amounts of the primary constituent (puerarin) of the kudzu root lyophilisate as well as a significant content of both hydrophilic (polysaccharides) and hydrophobic (lignans, saponins) ballast compounds. Obtaining minor isoflavone components from such a difficult plant matrix and their efficient purification was possible by using two combined preparative methods (CPC and FC). The proposed CPC procedure used a new methodological approach of adding an acidic modifier to the optimized biphasic solvent system EtOAc-EtOH-H2O/10:1:10 (*V/V/V*), which significantly improved separation efficiency, shortened analysis time, reduced the amount of solvents, and yielded a purified polar isoflavone fraction in the form of a sharp, easily eluted chromatographic band. Both preparative methods met the requirements of green chemistry, were simple, and could be recommended for larger scale isolation of mirificin and other related kudzu root isoflavones.

## Figures and Tables

**Figure 1 molecules-27-06227-f001:**
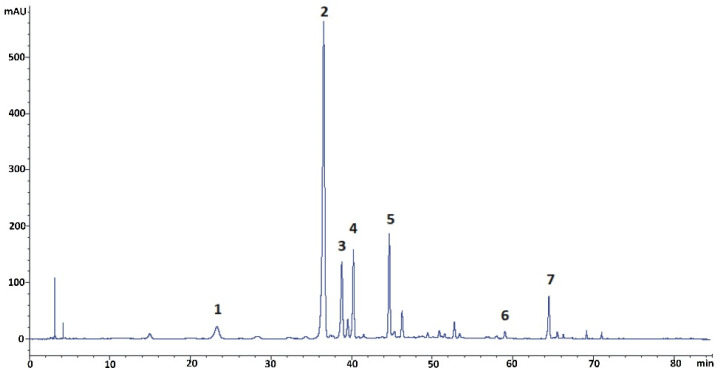
RP-LC/PDA chromatogram (registered at 260 nm) of isoflavonoids found in the kudzu root lyophilisate. Compounds: 1 = 3′-hydroxypuerarin; 2 = puerarin; 3 = 3′-methoxypuerarin; 4 = mirificin; 5 = daidzin; 6 = ononin; 7 = formononetin.

**Figure 2 molecules-27-06227-f002:**
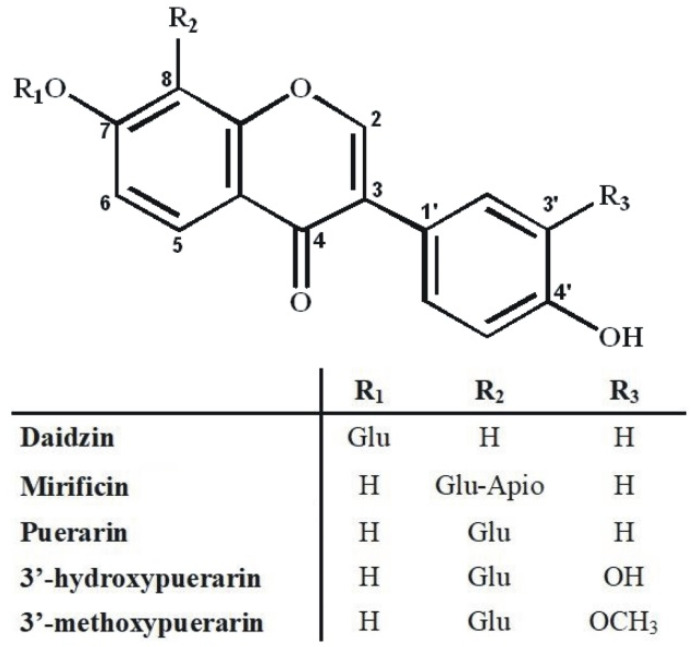
Molecular structure of polar isoflavone derivatives identified in the kudzu lyophilisate.

**Figure 3 molecules-27-06227-f003:**
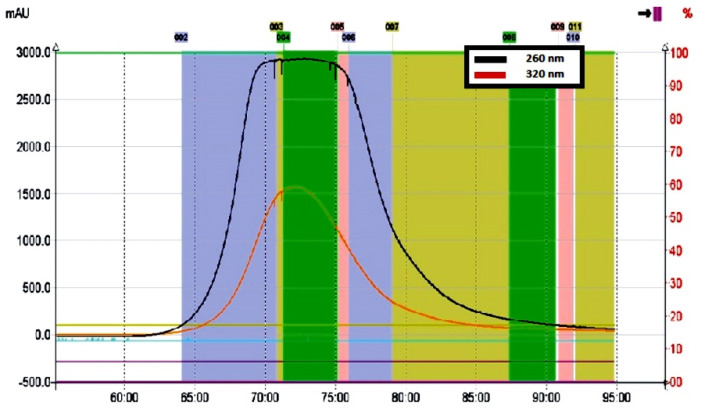
Chromatographic profile of polar isoflavone fraction recorded at 260 nm (black line) obtained from the kudzu root lyophilisate using the CPC method.

**Figure 4 molecules-27-06227-f004:**
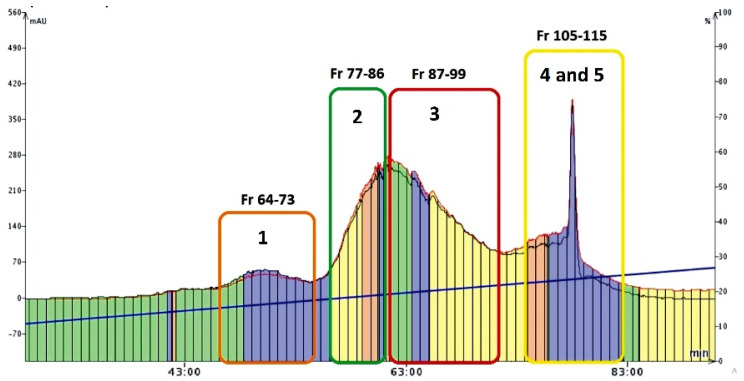
Chromatogram (recorded at 260 nm) showing the main fractions (eluates) with polar isoflavone glycosides (and compound numbers presented below) collected during the FC run.

**Figure 5 molecules-27-06227-f005:**
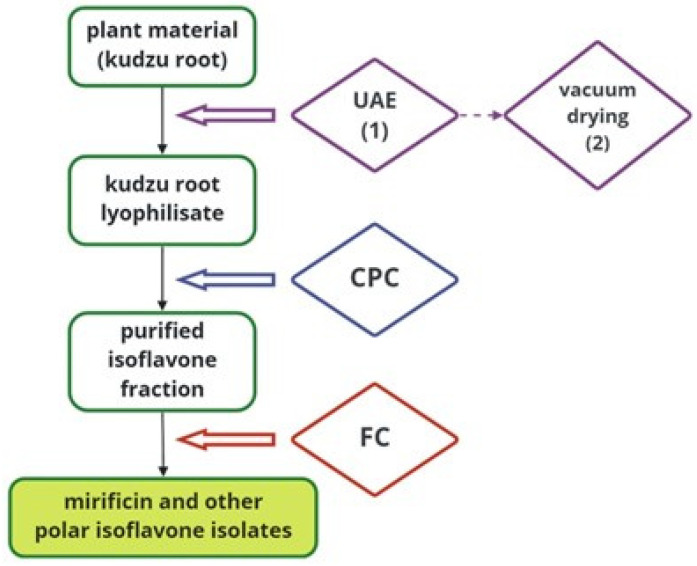
A block diagram illustrating the separation process of mirificin and related polar isoflavones using subsequent preparative techniques.

**Figure 6 molecules-27-06227-f006:**
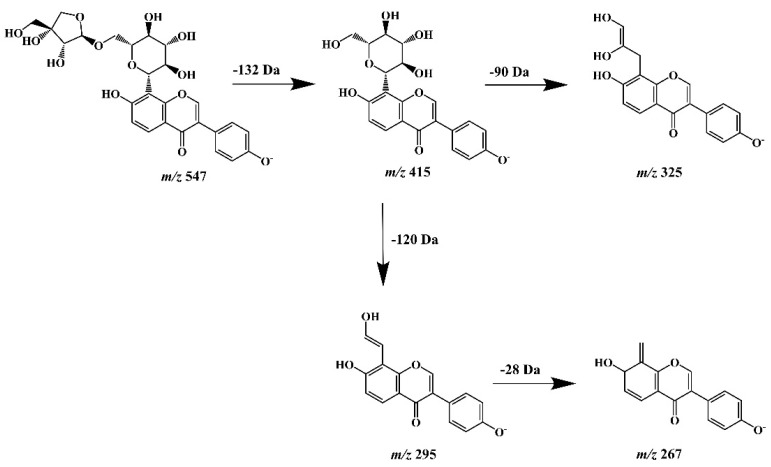
Hypothetical fragmentation pattern proposed to confirm the molecular structure of mirificin (*m*/*z* 547) based on the obtained mass spectrometric data.

**Table 1 molecules-27-06227-t001:** Two-phase solvent systems used in the optimization of the CPC procedure with the related partition coefficient (*K_D_*) values obtained for puerarin.

No	Biphasic Solvent System /Solvent Composition in Volume Proportions	*K_D_*
1.	EtOAc-BuOH-H_2_O/1:1:1.5	1.99
2.	EtOAc-BuOH-H_2_O/1:1:2	1.56
3.	EtOAc-BuOH-H_2_O/1:1:1	1.35
4.	EtOAc-EtOH-H_2_O/5:1:5	1.01
5.	EtOAc-EtOH-H_2_O/5:2:4	0.96
6.	EtOAc-EtOH-H_2_O/10:1:10 + 1% (*V/V*) CH_3_COOH	0.42
7.	EtOAc-EtOH-H_2_O/10:1:10 + 0.5% (*V/V*) CH_3_COOH	0.36

**Table 2 molecules-27-06227-t002:** The purity of polar kudzu root isoflavone isolates obtained in other studies realized using various counter-current preparative methods.

Compound Purity (%)					Method	Literature
M	D	P	3′-OHP	3′-MeOP		
-----	96,53	98,77	97.59	90.21	HPCCC	[19]
-----	> 95	>98	<90	>95	HSCCC	[30]
-----	-----	>99	-----	-----	CPC	[31]
-----	-----	<90	-----	-----	HSCCC	

*Abbreviations:* M- mirificin; D- daidzin; P- puerarin; 3′-OHP- 3′-hydroxypuerarin; 3′-MeOP- 3′-methoxypuerarin.

## Data Availability

Not applicable.

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
