# Peer review of "Isolation of Mirificin and Other Bioactive Isoflavone Glycosides from the Kudzu Root Lyophilisate Using Centrifugal Partition and Flash Chromatographic Techniques"

_molecules, 2022, doi:10.3390/molecules27196227_

Round 1

Reviewer 1 Report

author described the isolation method of mirificin and other isoflavones from kudzu root using centrifugal partition and flash chromatographic, this is an interesting topic but a lot of articles have been made on the isolation of isoflavone on kudzu root. my comment regarding the manuscript :

1. No purity data of the isolated compound has been written in the abstract 

2. other reasearch has been use on isolation of compound from P lobata in the introduction section line 91-96 as author said they proposed more efficient prepative isolation method

3. author should used better resolution of figures especially in figure 3

4. author should write the exact amount of the compound they have on line 303

5. in line 309, when author mentioned the purity of the individual isoflavone isolates, it has to be compared with other method to see the effectiveness of their method. I suggested author to use table of purity obtained from their method with other method. 

6. in the conclusions line 435, author mention that their method shortened analysis time, redued amount of solvents. how author conclude this when no comparation being made with other method? 

Author Response

Response to Reviewer 1

Dear Reviewer,  thank you so much for all your valuable remarks concerning our manuscript. Please find below our response (in black) to your comments (highlighted in blue).  

Author described the isolation method of mirificin and other isoflavones from kudzu root using centrifugal partition and flash chromatographic, this is an interesting topic but a lot of articles have been made on the isolation of isoflavone on kudzu root. My comment regarding the manuscript:

  1. No purity data of the isolated compound has been written in the abstract. 

The purity of mirificin has been addressed in the text (see the last line of the revised abstract).

  1. Other reasearch has been use on isolation of compound from P. lobata in the introduction section line 91-96 as author said they proposed more efficient preparative isolation method

In the Introduction part (lines 91-96), we emphasized that the CPC method we developed was more efficient because in the paper we cited [32], published by Sun and co-workers, the researchers isolated only the compound (puerarin) that was dominant in the isoflavone fraction, whereas in our study, in addition to puerarin, we isolated other polar isoflavones (first of all mirificin) with similar chemical structures, which were present in much lower concentrations than puerarin in the dry kudzu root extract.

  1. Author should used better resolution of figures especially in figure 3

Figure 3 and the other figures have been improved in terms of quality. New versions of each figure have been uploaded in JPEG format (400 dpi resolution).

  1. Author should write the exact amount of the compound they have on line 303

We included in the text the following sentence regarding the exact amounts of isoflavone isolates: “As regards to isoflavone isolates, in total, 68 mg of puerarin, 2 mg of 3’-hydroxypuerarin, 15 mg of 3’-methoxypuerarin, 18 mg of mirificin, and 20 mg of daidzin were obtained using the combined preparative procedures”

  1. In line 309, when author mentioned the purity of the individual isoflavone isolates, it has to be compared with other method to see the effectiveness of their method. I suggested author to use table of purity obtained from their method with other method. 

Following your suggestion, a new Table 2 was created (with a brief comment) showing the purity of isoflavone isolates obtained by other researchers.

  1. In the conclusions line 435, author mention that their method shortened analysis time, reduced amount of solvents. How author conclude this when no comparation being made with other method? 

We developed the original preparative procedure for isolation of mirificin using a hydrostatic CPC, followed by FC. There is only one report concerning the application of the two fast (below 25 min) CPC and HSCCC methods for the separation of polar isoflavone constituents of kudzu roots reported by Sun et al. [32]. However, the researchers focused on the isolation
of a single, predominant component of kudzu root extract (puerarin), whereas other minor polar isoflavones (including mirificin) were not separated and isolated. Moreover, aforementioned CPC and HSCCC methods were used in a micropreparative scale as a 10-mg samples of P. lobata extract were loaded into both chromatographic systems.

Writing in the conclusions about reducing the time of analysis, we referred to a cited recent study, conducted by Fu and co-workers [28]. The authors published (in 2015) a novel approach based on FC coupled to preparative high performance liquid chromatography (prep-HPLC) for the isolation of kudzu root components including mirificin. Although the purity of all isolated compounds ultimately exceeded 95%, the repeated short separation procedures using prep-HPLC were very tedious and took a total of about 300 min including FC isolation.

Reviewer 2 Report

The manuscript "Isolation of mirificin and other bioactive isoflavone glycosides from the kudzu root lyophilisate using centrifugal partition and flash chromatographic techniques" is well written and describes the development of an isolation procedure. There are only minor concerns that could improve the clarity of the paper.

1. lines 123-125, figure 2, figure 1 --number 1-7 were introduced as symbols for the investigated compounds but were used only in fig1 and in lines 116-118. The same numbers could be used in figure 2 (with the compounds of different order) and also in the remaining figures.  

2. too small font size in figure 1 axis description

3.  estimate concentrations of compounds 1-7 for figure 1 would be helpful

4. figure 4 - please add compounds number/symbols in respective fractions, enlarge font size (axes)

5. A block diagram illustrating the separation process would be helpful

Author Response

Response to Reviewer 2

Dear Reviewer, thank you so much for all the valuable remarks concerning our manuscript. Please find below our response (in black) to your comments (highlighted in blue).  

The manuscript "Isolation of mirificin and other bioactive isoflavone glycosides from the kudzu root lyophilisate using centrifugal partition and flash chromatographic techniques" is well written and describes the development of an isolation procedure. There are only minor concerns that could improve the clarity of the paper.

  1. lines 123-125, figure 2, figure 1 -number 1-7 were introduced as symbols for the investigated compounds but were used only in fig1 and in lines 116-118. The same numbers could be used in figure 2 (with the compounds of different order) and also in the remaining figures.  

Numbers were used only in Figure 1 due to lack of space on the chromatogram. However, we decided to leave the compound names (instead of the numbers) in Figure 2 to facilitate the reader's perception, while in Figure 4 we have added the compound numbers according to your suggestion.

  1. too small font size in figure 1 axis description

The font size on both axes of Figure 1 has been enlarged.

  1. estimate concentrations of compounds 1-7 for figure 1 would be helpful

Percentage concentrations of compounds 1-7 in kudzu root extract (that were shown as peaks in Figure 1)  have been added in the text of the revised manuscript (Section 2.1).

  1. figure 4 - please add compounds number/symbols in respective fractions, enlarge font size (axes)

This has been corrected. A new revised version of Figure 4 has been uploaded.

  1. A block diagram illustrating the separation process would be helpful

A new paragraph (2.4. Graphical presentation of preparative extraction and chromatographic procedures) in the Results and Discussion section was included, followed by a block diagram (Figure 5) illustrating the entire separation process.

Reviewer 3 Report

The manuscript entitled “Isolation of mirificin and other bioactive isoflavone glycosides from the kudzu root lyophilisate using centrifugal partition and flash chromatographic techniques” is an interesting contribution about the use of two chromatographic techniques to isolate mirificin. The introduction is well written, objective is clear, material and methods are reproducible, conclusions are supported by data. Only minor details are needed before accept the manuscript

Discussion

L286 Please use component instead “com-ponent”

L287 please use consistent instead “con-sistent”

Material and methods

Please indicate city and country of all instruments used in the experimental work

Author Response

Response to Reviewer 3

Dear Reviewer,  thank you so much for all your valuable remarks concerning our manuscript. Please find below our response (in black) to your comments (highlighted in blue).  

The manuscript entitled “Isolation of mirificin and other bioactive isoflavone glycosides from the kudzu root lyophilisate using centrifugal partition and flash chromatographic techniques” is an interesting contribution about the use of two chromatographic techniques to isolate mirificin. The introduction is well written, objective is clear, material and methods are reproducible, conclusions are supported by data. Only minor details are needed before accept the manuscript.

Discussion

L286 Please use component instead “com-ponent”

L287 please use consistent instead “con-sistent”

Dear Reviewer, thank you for your watchfulness. These typing mistakes have been corrected. 

Material and methods

Please indicate city and country of all instruments used in the experimental work

Missing information regarding the city and country of all instruments used in experiments has been completed in the Materials and Methods section.

Round 2

Reviewer 1 Report

The Authors already answered all my concerns, the article can be published in its present form